# Preventing Extremism with Extremists: A Double-Edged Sword? An Analysis of the Impact of Using Former Extremists in Danish Schools

**David Parker and Lasse Lindekilde \*** 

Department of Political Science, Aarhus University, Bartholins allé 7, 8000 C Aarhus, Denmark;
david.parker@ps.au.dk
\* Correspondence: lindekilde@ps.au.dk

**Abstract:** Governments across the West have invested significant resources in preventing radicalization, and strategies to prevent and counter violent extremism (P/CVE) are increasingly prominent within wider counter-terrorism policies. However, we know little about their effects, especially about projects that utilize former extremists to counter extremist narratives and increase critical thinking. Despite the prominence of interventions utilizing "formers", there are almost no systematic, scientific evaluations of these programs. The lack of evaluation is problematic given the recognized risks and negative effects of using formers to address other social issues, such as crime prevention. This paper presents findings from the largest study to date of the effects of using former extremists to prevent violent extremism. Based on a randomized controlled effect evaluation with 1931 Danish youths, it highlights significant successes, including reducing the perceived legitimacy of political violence, as well as negative effects, including a small decrease in political tolerance. Overall, the findings suggest a need for cost–benefit analyses of P/CVE initiatives, weighing the benefits against the risks.

**Keywords:** CVE; PVE; former extremists; radicalization; effect evaluation; extremism; formers

## 1. Introduction

The number of attempted and realized terrorist attacks in the West, in large part committed by home-grown actors [1,2], have increased in recent years [3,4]. Governments have invested significant resources and, given their controversial nature, substantial political capital on domestic initiatives designed to prevent or counter violent extremism (P/CVE) [5–7]. PVE has an anticipatory thrust and seeks to respond to personal or environmental conditions where violent extremism may flourish. CVE more directly addresses identified issues of extremism. Both seek to combat terrorism by using non-coercive methods (usually voluntary) to intervene before violence takes place, often seeking to enhance "resilience" to extremist ideas, enhance critical thinking, or create cognitive dissonance through counter-narratives [8–10]. Interventions frequently take the form of projects targeted at cohorts of people considered to be "at risk" or "vulnerable". One cohort regularly targeted is youth, based on assumptions about vulnerability of transition periods in life, a potentially weaker sense of identity, targeted efforts by extremist groups to mobilize young people online, and the prevalence of attacks committed by young people [11–13]. Consequently, interventions commonly take place in educational and youth settings, such as high schools, in the form of direct projects with young people [14] or training and support to help teachers identify and respond to suspected radicalization [15,16].

However, we know little about the impacts of these projects on young people. Few studies have analyzed the effectiveness of P/CVE initiatives empirically, and even fewer have applied research

designs that allow for causal inference [17]. In short, we do not know whether these initiatives meet the stated goals or whether they have unintended negative consequences, as some scholars suggest P/CVE activities and policies do [18,19]. This is especially the case for initiatives that use former extremists to discredit extremist ideologies, an increasingly common practice (see, e.g., the Against Violent Extremism Network, [20]). As early as 2014, the *Revised EU Strategy for Combating Radicalisation and Recruitment to Terrorism* urged greater engagement with former extremists and suggested that member states "collate and promote their testimonies" [21]. Several countries, including Northern Ireland, Spain, and Canada, have worked with former extremists or combatants with the aim of preventing radicalization of others [22–24]. Formers have also helped researchers to better understand the process of radicalization and shared their insights concerning how violent extremism can be countered [9,22]. This focus on former extremists stems, in large part, from the assumptions that "formers" will have unique credibility, expertise on the extremist ideology and effectiveness in de-glamorizing violence [21–28]. Despite the inherent risks involved in such initiatives, there are very few rigorous analyses of the effects.

The lack of rigorous evaluation of P/CVE intervention presents a clear research gap, with significant policy relevance. This paper begins to address this gap by testing the effects of a country-wide P/CVE intervention in Denmark. We provide findings from the largest empirical study to date of the impacts of using former extremists to prevent radicalization. The results of a randomized controlled effect evaluation with 1931 Danish young people, aged 13–20, exposed to a state-funded project utilizing former extremists, allow for a more nuanced understanding of the impacts. The paper tests the degree to which six project goals, formulated by the Danish authorities and organization delivering the formers project, are met. Amongst other findings, the analysis demonstrates effectiveness in lowering the perceived legitimacy of political violence, but also a reduction in political tolerance. The paper reviews existing literature on the evaluation of P/CVE and the use of former extremists, before outlining details of the intervention being evaluated. The methodology and data collection are detailed before a presentation of findings and a discussion of their implications for policy. Finally, the article identifies limitations and areas for future research. In addition to its contribution in terms of policy relevance, the paper expands on existing literatures in three ways. Firstly, it significantly advances the scarce research on the impacts of using former extremists to prevent radicalization. Secondly, it contributes more broadly to literature that evaluates the effectiveness of P/CVE initiatives, particularly as one of the first evaluations to employ randomized controlled trial designs to allow for controlled testing of effects. Finally, it adds to research that evaluates the impact of using formers in tackling social problems more generally.

## 2. Evaluating P/CVE and Use of Formers

The process of radicalization has been widely explored [29–33]. We understand radicalization as the process by which an individual comes to see violent extremism as a viable action alternative [29]. The process has been explained with reference to individual level factors, such as experiences of self-uncertainty and personality traits, social networks, and group-dynamics at an inter-personal level, as well as macro level factors, such as neighborhood characteristics and political ideologies [30–33]. There are also multiple critiques of P/CVE strategies, even if many are built primarily on theory rather than substantive empirical data [34,35]. However, when it comes to identifying the actual effects of specific P/CVE initiatives there are two core problems. The first is a dearth of robust evaluations generally and, secondly, barely any quantitative evaluations that utilize sophisticated methodologies to identify causality. A 2014 synthesis by van Hemert et al. of the state of the art in evaluating the effectiveness of P/CVE interventions found that only 16% of published data had been peer reviewed, 64% were assessed to be low quality, and only 0.8% used experimental methods. The authors concluded that "hardly any empirically based evidence of counter-radicalization interventions exists [ . . . ] there is a significant lack of empirical tests of propositions and assumptions" [17]. Similarly, a 2015 review by Feddes and Gallucci found that "hardly any empirically based evidence of preventive or

de-radicalisation interventions exists" and that most studies of CVE have "no explicit reference to theory and no empirical quantitative or qualitative data was reported" [36].

A few studies have analyzed the effectiveness of radicalization prevention initiatives empirically in recent years [37]. In 2015, Feddes, Mann, and Doosje reported on a longitudinal evaluation of a resilience training program with Muslim adolescents and young adults with migrant backgrounds. The authors demonstrated successes alongside unexpected effects. Positive attitudes towards violence were significantly reduced, but there was a marginal increase in participant narcissism [38]. This is a significant finding, as it highlights the mixed effects of P/CVE interventions, moving us beyond a frame of "what works" to a fuller consideration of the wider effects. In a 2016 analysis of the effects of a US-based CVE community program, Williams, Horgan, and Evans found that 12 out of 14 objectives in relation to CVE outcomes concerning attitudes, behaviors, relationships, and knowledge had been met. These included learning about other cultures, feeling part of something bigger than oneself, and feeling accepted [39].

Despite these recent advances, the broad conclusions about the lack of effect evaluations remain relevant. In a 2016 analysis, Mastroe and Szmania identified 43 published, peer-reviewed assessments of P/CVE programs that included empirics. The majority "reported descriptive results, while only twenty-four studies provided data that could be broadly categorized as correlational findings of program effectiveness". The authors found no evaluations based on experimental designs employing randomized controlled effect assessment [40]. Likewise, a 2017 review by Gielen found that only 14 academic studies addressing CVE evaluation can be considered "effect evaluations" [37]. Scholars thus continue to emphasize the lack of knowledge about the effects of P/CVE initiatives [41,42]. Of course, effective evaluation of complex projects designed to prevent behaviors and shape attitudes is challenging [43,44]. Nevertheless, the need to improve evaluation is essential. Alex Schmid has argued that, in the context of ever increasing numbers of projects seeking to counter radicalization, evaluation "should have one of the highest priorities", because without systematic and rigorous evaluations, "no real progress towards more promising practices can be made" [45]. Similarly, Koehler described the question of how to evaluate deradicalization and prevention projects as "the most pressing and important issue in the field" [46]. The lack of scientific analyses of primary data to isolate the effects of specific P/CVE programs is clearly an important gap in knowledge for both researchers and policy-makers.

Despite investment in formers projects, "there has been little research evaluating the effectiveness of formers in CVE initiatives" [21,40,43]. This is troubling for at least two reasons. Firstly, research analyzing the use of formers in preventing other damaging social behaviors has found mixed results. This includes cases where the intervention is more harmful than doing nothing. For example, some "Scared Straight" programs (projects with former prisoners designed to prevent crime and delinquency amongst youths) may have actually contributed to increased levels of crime and delinquency. This is, in part, because some youth participants see prison as an opportunity for friends and community rather than a deterrent [47,48]. Scholars have noted that similar risks apply when former extremists are employed to counter extremism. Risks include providing implicit justification for violence [24], supplying a platform for extremists whose world views have not genuinely changed [21], expressing intolerant views despite renouncement of violence [49], or reinforcing the formers' power and influence, which may not be welcomed by the community [25]. Secondly, such projects have the same aims as many other counter-radicalization initiatives: to promote critical thinking and increase resilience to extremist narratives. As noted, there are few evaluations of the relative effects of these approaches, particularly of unintended consequences.

Finally, some literature suggests that using a theater format (as the project evaluated in this paper does) to address challenging or controversial topics can be particularly effective. In the theater project format, the formers are on stage/at the front of the room and deliver monologues they have prepared in cooperation with a professional theater company, to an audience of students. The effectiveness of so-called "entertainment-education" has been outlined in a range of studies, highlighting benefits in

terms of increasing knowledge, creating favorable attitudes, and influencing overt prosocial and health behaviors [50,51]. Indeed, studies suggest that working with sensitive topics in a theater format can act as a catalyst for more thoughtful conversations that correspond to increased empathy, understanding, and more prosocial attitudes [52,53]. A recent study by Flesner, Larsson, and Säljö indicated that a theater-style approach to issues concerning extremism and terrorism could have similar effects. Employing a qualitative analysis to assess the impacts of two plays delivered to upper secondary school students in Sweden, one focused on migration and the other on radicalization and jihadism, the authors identified a range of effects to confirm that the plays could affect viewpoints and behaviors. These included positive effects, particularly a majority consensus that religious, violence-approving extremism is "bad". The authors also found that real-life cases were more impactful than hypothetical cases delivered by actors, which suggests that the use of real-life former extremists could be especially effective. However, there were also negative or unexpected effects. For instance, several students felt that *The Jihadist* provided excuses for those who had undertaken illegitimate actions, and a few suggested after the performance that Islam should be abolished [14]. Such findings relate to broader studies that highlight the potential for unintended effects of P/CVE projects.

## 3. Background to the Danish Formers Project

Although Denmark has had relatively few historic encounters with terrorism, authorities have been on high alert since the 2005 Danish cartoon crisis, in which images depicting the Prophet Muhammad were published in the Danish newspaper *Jyllands-Posten*, causing the most severe Danish foreign policy crisis since the Second World War, with multiple attacks on Danish representations abroad and massive Muslim mobilization around the world [54]. Since that controversy, there has been one successful attack (the 2015 Copenhagen shooting) and at least six thwarted plots. Concerns were exacerbated by an estimated 150 Danish citizens travelling to Syria or Iraq since 2012, at least half of whom have since returned to Denmark [55]. In addition to fears about Islamist extremism, there are at least 10 right-wing or radical left-wing groups in Denmark, and hate crimes against immigrants are increasing [56]. In response to these developments, the authorities have developed a relatively sophisticated anti-radicalization infrastructure to prevent or counter radicalization. The most important aspect is the Info-House framework, a format for multi-agency cooperation to assess and support individuals deemed to be most at risk of radicalization. All 12 police districts have established an Info-House, and local and national public bodies also fund a range of projects designed to address the risks of radicalization [55,56].

One of the most significant recent P/CVE projects is a state-funded initiative that organizes former extremists to visit schools and youth centers across Denmark to talk about their experiences. Danish authorities asked the authors, in summer 2018, to evaluate the effectiveness of the initiative. The project has a pool of approximately 12 former extremists, and for most performances, five formers are selected based on their relevance to the specific audience. The diverse set of formers, from across the ideological spectrum (e.g., Islamist, former Nazi, far-left autonomous), talk on stage about their experiences. The formers in the project vary in terms of how long they were exposed to extremism and were part of radical groups, but they all have personal experiences of engaging in or supporting violent extremism. The monologues emphasize the negative impact extremism and violent extremism had on their lives, and they describe how they got exposed to extremist ideologies and how it made them think and act. In this way, the intervention tries to address and shed light on mechanisms of radicalization at an interpersonal level of exposure and recruitment and at the level of extremist worldviews and ideologies.

The monologues (each former speaks for approximately ten minutes) were designed in collaboration with a professional theater company. Following the monologues, students break into smaller groups for a 30-min workshop where the former extremists facilitate focused discussion. The project has run previously in Denmark, and an internal evaluation conducted by the theater company showed positive effects. However, these were solely self-reported satisfaction and gains, and Danish authorities were keen for external assessment based on more rigorous academic evaluation

techniques. The theater company describes the overarching goal of the project as increasing "democratic confidence", and the project is funded with the aim of increasing societal resilience to radicalization. Project attendees were young people aged 13-20, living in Denmark. The performances took place at schools, youth centers and local theaters between October 2018 and February 2019. All performances were observed by a teacher or a relevant member of staff.

## 4. Methodology and Data

### 4.1. Evaluation Standard

As in Schuurman and Bakker's evaluation of a Dutch initiative to reintegrate Islamist extremists [57], the authors first sought to identify the project's underlying aims and assumptions and use those as the evaluation standard. Following a series of discussions with the funders and representatives of the theater company, four specific goals were identified that the funders expected the project to deliver: 1) increase political tolerance, 2) increase political efficacy, 3) improve ability to recognize extremist recruitment tactics, and 4) increase confidence in knowing what to do if exposed to extremism. In addition, the authors chose to assess one further outcome: 5) whether the project reduced perceived legitimacy of political violence. The measurement of acceptance of political violence was included, as that is one of the overarching aims of the state's P/CVE work. Finally, the funders also requested information to 6) indicate participant satisfaction with the project.

### 4.2. Research Design

To investigate the impacts of the project against the aims formulated by the funders and organization delivering the project, the authors designed an effect evaluation using a survey experiment. Attendees ($n = 2156$) were randomly assigned to one of two conditions: 1) no exposure to the project (control), and 2) exposure to the project (treatment). Participants in the control condition answered all questions operationalizing the aims of the project prior to the performance, and in the treatment condition, attendees answered the same questions at the conclusion of the workshop after the performance. To evaluate the effectiveness of the initiative in reducing perceived legitimacy of political violence, the authors embedded a list experiment in the survey. It was assessed that this would be a more valid measure of attendee support for political violence than direct questions. A list experiment (or the item-count technique) utilizes an indirect questioning technique to reduce dishonest or evasive responses on sensitive topics [58]. Using these methods, the study attempted to identify the causal effect of the intervention on support for political violence.

In order to enhance the understanding of baseline perceptions, the core survey questions were completed by a separate cohort ($n = 658$) of Danish high school students, who were at no point exposed to the formers' performance. The sample was representative of gender and geographical location across Denmark. This sample provided a useful point of comparison for the schools that participated in the formers project—some of which were specifically identified (by the authorities) as having student cohorts where P/CVE initiatives were especially relevant. Thus, the sample of students who were exposed to the formers project predominantly came from inner-city schools and schools/youth centers in deprived neighborhoods, but also included students from schools in the province and non-deprived neighborhoods. The baseline sample was closer to a representative sample of Danish school students.

### 4.3. Data Collection

The evaluation process was designed by the authors, who attended several early shows to ensure that data were collected as outlined in a written guide provided to the theater company administering the project. Survey responses were collected between October 2018 and February 2019. In total, the evaluation was conducted at 32 out of 40 performances. For the remaining eight performances, audiences were deemed to be too large (e.g., over 200) for the evaluation to be properly managed by the few theater company staff in attendance. Schools or youth centers signed up for the performance,

and attendance was mandatory for students. As mentioned, an extra effort was made to promote the project to schools with a student population deemed to have been at risk from exposure to extremism (predominantly schools in the larger cities), but participation was offered and open to all schools. All schools and youth centers that expressed interest in the performance received an invitation. Before beginning the survey, participants were informed about the purpose of the study. No deception was employed, and participants were informed about how their data would be stored and used. Informed consent was collected from attendees. Where the consent box was not ticked, the data were excluded from the study.

The data collection process centered on a pen-and-paper survey distributed to attendees and overseen by the theater company. Randomization of participants to condition (control and treatment) was obtained by randomly providing all attendees with one of two different "evaluation envelopes" as they entered the performance. Before the performance began, all attendees were asked to open the envelope, which contained an initial set of questions (on paper, pen provided) and a smaller, sealed envelope that participants were instructed not to open. Participants were instructed not to talk to each other while filling out the survey. For half the attendees (control condition), the initial questions addressed the aims of the project and background questions about previous exposure to extremism. For the other half (treatment condition), the questions focused on demographic information, including age, gender, postcode, and questions about previous exposure to extremism. After the conclusion of the performance and workshop, attendees were asked to open the sealed envelope and answer the questions within. For the attendees who had initially answered questions concerning the project's goals, the second set of questions were the demographic questions that treatment condition participants completed prior to the performance. For those who had initially answered the demographic questions, the smaller envelope contained the questions operationalizing the project aims. Control group participants thus only answered questions measuring the factors the project aimed to influence (e.g., political tolerance) before the performance began. The attendees in the treatment condition answered the same questions after the conclusion of the performance. Comparison of the control and treatment condition thus allows for isolating the causal impact of the project. Attendees typically spent between 10 and 15 min completing the survey both before and after the intervention, which lasted approximately 90 min. This method permitted meaningful analysis of the immediate effects of the intervention, but more longitudinal research will be required to assess the degree to which these effects last over a longer period.

### 4.4. Sample Characteristics

The survey was completed by 2156 participants. After data cleaning (excluding participants who failed to tick the informed consent form, who straight-lined all responses, or who were adults/teachers), the final $N$ was 1931, with participants aged 13–20. Of these, 976 were in the control condition (50.5%) and 955 were in the treatment condition (49.5%). A total of 50% of participants were female, 44% male, and 6% preferred not to answer. A total of 538 participants were aged 13–14 (33%), 717 aged 15–16 (40%), 392 aged 17–18 (22%), and 105 aged 19–20 (5%). Mean age was 15.5. A total of 64% of participants were enrolled in public primary schools ($n = 1.188$), 22% in secondary youth education (STX, HF, HTX, HHX) ($n = 396$), and the rest in private primary schools (73), vocational schools (111), higher education (24), or other (55). Participants were from across Denmark, with a majority (1207) from Copenhagen and surrounding areas, and 724 from the provinces.

A comparison of the control and treatment groups on demographic variables confirmed that the randomization worked in creating similar groups. Thus, a set of t-tests showed that there was no significant difference in age for the control (M = 15.44; SD = 1.64) and treatment group (M = 15.54, SD = 1.68; t (1796) $= -1.28$, $p = 0.19$), in proportions of those residing in cities (64% versus 62%; z $= -0.84$, $p = 0.40$), females (55% versus 52%; z = 1.36, $p = 0.17$), or students in primary school (66% versus 64%, z $= -0.78$, $p = 0.43$). Likewise, there was no difference in self-estimated benefit from

the performance in the control (M = 3.89, SD = 0.92) and treatment condition (M = 3.92, SD = 0.91; t (1757) = −0.86, $p$ = 0.38).

## 4.5. Operationalization: Constructs and Measures

To assess the effect of the performance against the evaluation standard, we included in the survey measures of political tolerance, political efficacy, ability to recognize extremist recruitment tactics, confidence in knowing what to do if exposed to extremism, perceived legitimacy of violence, and participant satisfaction with performance. As control variables we included measures of gender, age, location, and previous exposure to extremism.

Political tolerance involves accepting the political rights of others, such as freedom of speech, even for groups with which one disagrees or fears [59]. Tolerance of difference has been identified as a potentially protective factor against violent extremism, curbing "us and them" thinking and heightening acceptance of the unfamiliar. Efforts to enhance tolerance (or reduce intolerance) feature in P/CVE efforts across a range of contexts [60–62]. Political tolerance was measured using three items ($\alpha$ = 0.79), based on measures used previously by Peffley, Knigge, and Hurwitz [63], that capture tolerance of a political/religious group disliked by the participant to make a speech in their city (M = 3.94, SD = 1.09), hold a meeting in their neighborhood (M = 3.43, SD = 1.14), and to use Facebook to recruit to their group (M = 3.51, SD = 1.23).

Political efficacy refers to people's trust in government and belief that they understand and can influence politics. Studies have shown that higher levels of political efficacy may not only reduce the psychological factors associated with support for radicalization to violence [64], but also increase willingness to participate in counter-extremism activities in specific contexts [65]. The measures selected focused on internal efficacy, meaning an "individual's self-perceptions that they are capable of understanding politics and competent enough to participate in political acts" [66]. Two commonly used items were utilized [67,68], namely perceptions concerning the degree to which: (i) politics and government seem so complicated that someone like the participant cannot really understand what is going on (M = 3.62, SD = 1.15), and (ii) people like the participant do not have any say about what the government does (M = 2.65, SD = 1.33). A combined scale showed a relatively low internal consistency ($\alpha$ = 0.35), so we looked at the items separately as well as combined in the analysis. Both political tolerance and political efficacy were measured on a five-point Likert scale. Options were "strongly agree", coded as a score of (1), "tend to agree" (2), "neither agree nor disagree" (3), "tend to disagree" (4), and "strongly disagree" (5).

To measure the impact of the project on participant ability to recognize extremist recruitment methods, participants were presented with eight possible methods and asked to assess on a five-point scale how likely extremist groups were to use each method. Some of the options were genuine methods drawn from the radicalization literature, such as offering excitement and status [69,70]. Others were unassociated with extremist recruitment, such as providing detailed, fact-based arguments, and giving people time to research the issues independently. The internal consistency of this scale ($\alpha$) was 0.70. Research has shown that there are a range of online and offline pathways to membership of extremist groups [71,72], and enhancing the ability of youth to recognize recruitment methods is therefore important for many P/CVE initiatives.

The authors were unable to identify a suitable existing measure of confidence in responding to extremism and therefore designed three items ($\alpha$ = 0.60) based on the project content and discussions with P/CVE practitioners. Participants were asked to indicate the degree to which they agreed that they: (i) could recognize extremist ideas if they came across them (M = 3.75, SD = 1.00), (ii) would know what to do if they heard extremist conversations in school/everyday life (M = 3.41; SD = 1.07), and (iii) would know where to get help if an extremist was trying to exploit them or a friend (M = 3.36, SD = 1.25). Options were "strongly agree", coded as a score of (1), "tend to agree" (2), "neither agree nor disagree" (3), "tend to disagree" (4), and "strongly disagree" (5). The capacity to recognize extremism, and to understand how to access support, is important not only for enhancing protective

factors around the individual but also for wider approaches to extremism, which increasingly ask the public to report suspicious behaviors or concerns about radicalization to the authorities [73–75].

The list experiment to measure the degree to which the project affected perceived legitimacy of political violence randomly assigned participants (across both control and treatment condition) to assess how many of four or five statements they agreed with. Inspired by Dinesen and Sønderskov [76], the authors included one statement that they expected most people would agree with ("I like watching movies"), one statement they expected most people would not agree with ("I want to work as a garbage collector"), two statements they expected people to disagree about ("In schools they ought to teach more music and dancing" and "In schools they ought to teach more religious values"), and in the five-item condition also the critical item ("It can be justified to use violence against public authorities or politicians"). We chose this critical item (the technique allows for only one) because the Danish authorities were particularly concerned about direct threats to democracy. Participants were instructed to indicate how many rather than which specific statements they agreed with. The order of items was randomized. The mean difference between the four- and five-item conditions was interpreted as the percentual acceptance of the critical item.

A single measure of participant satisfaction with the project was used, asking participants to select the degree of benefit they felt from hearing about the formers' experiences (M = 3.90, SD = 0.91). There were five options: (1) "no benefit", (2) "small benefit", (3) "average benefit", (4) "large benefit", and (5) "very large benefit".

Control variables: standard items were used to capture gender and age. Location was measured by having participants report postal code of residency. Previous exposure to extremism was measured using three items ($\alpha = 0.61$) focused on participant experiences of: someone they know "liking extremist content on the internet" (M = 0.56, SD = 0.76), extremist propaganda in their school or neighborhood (M = 0.34, SD = 0.64) and, finally, concerns about extremism vis-a-vis someone they know (M = 0.41, SD = 0.67). Participants were asked whether they had experienced these situations (1) never, (2) once, or (3) more than once.

### 4.6. Data Analysis

To evaluate the identified aims against the collected data, the authors ran a set of linear regressions with political tolerance, political efficacy, ability to recognize extremist tactics, and perceived legitimacy of political violence as outcome variables. The main predictor was the dummy-coded condition variable (before/after show). In all models, we included gender, age, location, previous exposure to extremism, and self-reported benefit from show as controls. In an additional model, we regressed self-reported benefit from show on previous exposure to extremism and the control variables to estimate who benefited the most from the show. The list experiment was analyzed using a set of t-tests of the differences between mean number of items agreed with across conditions.

## 5. Results

The following section outlines the extent to which the Danish state-funded former extremist project met the six agreed goals, including assessing attendees' self-reported benefit.

### 5.1. Goal 1: Increase Political Tolerance

In contrast with the project's aim, Table 1 shows a statistically significant decrease in political tolerance following exposure to the performance and workshop ($b = -0.05$, SE = 0.01, $p = 0.000$).

**Table 1.** Effect of show on political tolerance.

|  | Model I | Model II |
|---|---|---|
| Gender (reference = male) | 0.02 (0.01) [*] | 0.01 (0.01) |
| Age | 0.00 (0.00) | 0.00 (0.00) |
| Location (reference = city) | 0.02 (0.01) | 0.02 (0.01) [*] |
| Exposure to extremism | 0.03 (0.02) | 0.03 (0.02) |
| Self-reported benefit | 0.01 (0.00) [**] | 0.02 (0.00) [**] |
| Show (reference = before) |  | −0.05 (0.01) [***] |
| Constant | 0.55 (0.06) | 0.57 (0.06) |
| Adj $R^2$ | 0.001 | 0.018 |

Note: $N$ = 1680. Political tolerance is scaled 0-1. Show is coded 0 = before show; 1 = after show. Exposure to extremism is scaled 0-1. Standard errors in parentheses. [†] $p < 0.10$, [*] $p < 0.05$, [**] $p < 0.01$, [***] $p < 0.001$.

This corresponds to a 5.2% decrease in political tolerance. The decrease holds across age. When participants were split into two age groups—13–16 (primary school) and 17–20 (secondary school)—a statistically significant drop in political tolerance was apparent for both groups. There was no significant difference in the drop in political tolerance between participants from large cities and from the provinces. Self-reported benefit of the show was positively correlated with political tolerance ($b$ = 0.02, SE = 0.00, $p$ = 0.003). The decrease is particularly notable, as the baseline political tolerance (control condition) was already substantially lower than the findings from the survey conducted by the authors with 658 Danish high school students (nationally representative on gender and region), who averaged a mean of 0.77 using the same measures, compared to 0.68 in the evaluation sample. The funders and theater company had hoped that hearing about how embracing previously hated groups (e.g., the former Palestinian Islamist who now has an Israeli kick-boxing coach and friend) was the route to mainstream society and a better life for the formers, would increase attendee tolerance of groups different from one's own. However, it seems that stories of how extremist groups initially influenced the formers in a negative way, (mis)using political liberties to recruit and distribute their extremist narratives, was a more significant driver of changes in political tolerance. Confronted with the formers' different stories about the negative consequences of exposure to and involvement with extremist groups, attendees may have come to think that groups that they and most people disagree with should not be tolerated and left free to maneuver.

*5.2. Goal 2: Increase Political Efficacy*

The analysis found that the initiative had no statistically significant effect on political efficacy.

Both for the two political efficacy items separately and combined, there was a small reduction in political efficacy, although none were statistically significant. The authors considered the possibility that the finding could be influenced by political opportunities, primarily the association between political efficacy and voting rights (18 years in Denmark). This interpretation is supported by the fact that the only robust finding in Table 2 is that political efficacy increases with age (shown as a negative correlation with the view that "politics is too complicated" ($b$ = −0.00, SE = 0.00, $p$ = 0.044) and "I have no say in what the government does" ($b$ = −0.01, SE = 0.00, $p$ = 0.006)). However, an analysis of impact on participants aged 13–16 and 17–20 showed no difference. Findings from the supplementary survey with 658 Danish high school students found that, as with political tolerance, baseline political efficacy was lower amongst participants in the formers project than in the Danish high school student sample.

**Table 2.** Effect of show on political efficacy.

|  | Politics too Complicated | No say in what Govern. Does | Both Items Combined |
|---|---|---|---|
| Gender (reference = male) | 0.08 (0.01) *** | −0.02 (0.01) | 0.02 (0.01) * |
| Age | −0.00 (0.00) * | −0.01 (0.00) ** | −0.01 (0.00) ** |
| Location (reference = city) | −0.00 (0.01) | 0.03 (0.01) * | 0.01 (0.01) |
| Exposure to extremism | 0.04 (0.02) | 0.06 (0.03) * | 0.05 (0.02) * |
| Self-reported benefit | 0.00 (0.00) | 0.01 (0.00) | 0.00 (0.00) |
| Show (reference = before) | −0.00 (0.01) | −0.00 (0.01) | −0.00 (0.01) |
| Constant | 0.72 (0.07) | 0.57 (0.08) | 0.65 (0.06) |
| Adj $R^2$ | 0.023 | 0.005 | 0.001 |

Note: $N = 1680$. Both political efficacy items and combined scale are scaled 0-1. Show is coded 0 = before show; 1 = after show. Exposure to extremism is scaled 0-1. Standard errors in parentheses. $^\dagger$ $p < 0.10$, $^*$ $p < 0.05$, $^{**}$ $p < 0.01$, $^{***}$ $p < 0.001$.

### 5.3. Goal 3: Increase Ability to Recognize Extremist Groups' Recruitment Techniques

In line with the stated goals, Table 3 shows that the project led to a small increase in participants' ability to correctly recognize recruitment techniques frequently used by extremist groups ($b = 0.02$, SE = 0.00, $p = 0.004$).

**Table 3.** Effect of show on ability to recognize extremist groups' recruitment techniques.

|  | Model I | Model II |
|---|---|---|
| Gender (reference = male) | −0.00 (0.00) | −0.00 (0.00) |
| Age | 0.01 (0.00) *** | 0.01 (0.00) *** |
| Location (reference = city) | 0.02 (0.00) * | 0.02 (0.00) * |
| Exposure to extremism | −0.03 (0.01) * | −0.03 (0.01) * |
| Self-reported benefit | 0.03 (0.00) *** | 0.03 (0.00) *** |
| Show (reference = before) |  | 0.02 (0.00) ** |
| Constant | 0.42 (0.04) | 0.41 (0.04) |
| Adj $R^2$ | 0.055 | 0.060 |

Note: $N = 1680$. Ability to recognize extremist groups' recruitment strategies is scaled 0-1. Show is coded 0 = before show; 1 = after show. Exposure to extremism is scaled 0-1. Standard errors in parentheses. $^\dagger$ $p < 0.10$, $^*$ $p < 0.05$, $^{**}$ $p < 0.01$, $^{***}$ $p < 0.001$.

This corresponds to a 2% increase in ability to recognize extremist recruitment methods. Younger participants drove the increase. Whilst there was an increase for both 13–16-year-olds and 17–20-year-olds, this increase was only statistically significant for the younger cohort, who had a lower baseline. In terms of geographical context, the increase was significant for participants living in both cities and provinces. A little surprisingly, the participants from the provinces started with a higher mean baseline (2.85 versus 2.72, scale 1–5). Exposure to extremism was a negative predictor of ability to recognize extremist groups' recruitment techniques ($b = −0.03$, SE = 0.01, $p = 0.016$), indicating the need to target the performance to students with previous exposure to extremism.

### 5.4. Goal 4: Increase Confidence in Responding to Extremism

Table 4 highlights considerable success in increasing participant confidence in recognizing and responding to extremism ($b = 0.26$, SE = 0.03, $p = 0.000$).

**Table 4.** Effect of show on confidence in responding to extremism.

|  | Model I | Model II |
|---|---|---|
| Gender (reference = male) | −0.21 (0.04) *** | −0.20 (0.04) *** |
| Age | −0.04 (0.01) *** | −0.04 (0.01) *** |
| Location (reference = city) | −0.12 (0.04) ** | −0.12 (0.04) ** |
| Exposure to extremism | 0.26 (0.07) ** | 0.25 (0.07) ** |
| Self-reported benefit | 0.09 (0.02) *** | 0.08 (0.02) *** |
| Show (reference = before) |  | 0.26 (0.03) *** |
| Constant | 3.92 (0.20) | 3.82 (0.20) |
| Adj $R^2$ | 0.047 | 0.073 |

Note: $N$ = 1680. Ability to recognize extremist tactics is scaled 0-1. Show is coded 0 = before show; 1 = after show. Exposure to extremism is scaled 0-1. Standard errors in parentheses. [†] $p < 0.10$, [*] $p < 0.05$, [**] $p < 0.01$, [***] $p < 0.001$.

The changes correspond to a 3% increase in perceived ability to recognize extremist ideas, a 4% increase in perceived ability to know what to do if exposed to extremism, and a 10% increase in perceived ability to know where to get help. These effects were not age-specific, while age was negatively correlated with confidence in responding to extremism ($b = -0.04$, SE = 0.01, $p = 0.000$). There were some important differences in terms of geographic context. Participants in the largest cities reported greater baseline confidence than participants in the provinces (3.58 versus 3.38, scale 1–5), and being from the province was a negative predictor of confidence ($b = -0.12$, SE = 0.04, $p = 0.003$). This corresponds with participants in the largest cities having more experience with extremism (see below). In terms of effects of the performance, the increase in recognizing extremist ideas was driven by participants in the provinces. Increases in knowing what to do and where to get help were significant for participants from both the cities and the provinces. Previous exposure to extremism was a strong positive predictor of confidence in responding ($b = 0.25$, SE = 0.07, $p = 0.000$).

The positive effects of the performance on confidence can also be seen in comparison to the 658 Danish high school students who answered the same questions in the baseline survey. These students had a higher baseline for confidence in recognizing extremist ideas (3.87). However, the average respondent seeing the formers' performance was more confident in knowing what to do and how to get help if exposed to extremism. This increased after exposure to the show, meaning that individuals who saw the performance were significantly more confident in knowing how to respond if exposed to extremism than the students in the baseline survey.

*5.5. Goal 5: Identify Impact on Perceived Legitimacy of Political Violence*

Whilst reducing the perceived legitimacy of political violence was not a stated goal of the theater company, it is one of the primary aims of P/CVE programs based on the assumption that it can be an important factor in radicalization to violent extremism [26]. Indeed, the findings from the list experiment indicate that the project was successful in lowering the perceived legitimacy of political violence against public authorities or politicians.

Table 5 shows the average number of items agreed to by participants across the control and treatment condition by whether they saw four or five items in the list experiment. The average difference between the four-item and the five-item condition is interpreted as the percentual agreement with the critical item—here, political violence against public authorities or politicians [58]. The results show that acceptance of political violence dropped from 33.5% before the intervention to 8.9% after the show—indicating a reduction of an estimated 24.6% of participants who view political violence as legitimate. The fall was particularly large amongst younger participants (drop of 34.3%), aged 13–16, and those living in cities compared to the provinces (drop of 23.3% versus 12.7%). The baseline survey with 658 Danish high school students found a baseline of 5% support of violence against public authorities or politicians based on the same list experiment. When students in this study were asked directly about political violence, only 4% agreed. Whilst the support was still higher following

the formers project, it was much closer to the average scores of the baseline survey than prior to the intervention.

**Table 5.** Effect of show on perceived legitimacy of political violence. Number of items agreed to by conditions.

|  | Four-Item Condition | Five-Item Condition (Critical Item = "Violence against Public Authorities or Politicians") | Contrast |
|---|---|---|---|
| Before show | 1.81 | 2.15 | 0.335 *** |
| After show | 1.94 | 2.03 | 0.089 |

Note: Results as average number of items agreed to. The contrast between the four-item and five-item condition is interpreted as the percentual agreement with the critical item. $^*p < 0.05$, $^{**}p < 0.01$, $^{***}p < 0.001$

### 5.6. Goal 6: Measure Participant Satisfaction with the Project

Many scholars argue that self-reported assessments are an unsatisfactory means to measure the effectiveness of an intervention [77]. Nevertheless, it is interesting, and potentially indicative, to observe participants' own assessments of the project. On a scale from 1 (no benefit) to 5 (very significant benefit), the self-reported mean benefit was 3.9. Of the 1759 participants who completed this question, only 2% (32) reported no benefit. A total of 3% (55) reported a "small benefit", and 27% (471) an "average benefit". The mode was "large benefit", with 39% (685) selecting this option. Finally, 29% (515) recorded the maximum benefit of "very significant benefit". Such findings should not be used as the basis for firm conclusions on the effect or relevance of a project. Nevertheless, as outlined above, researchers have identified a clear correlation between the entertainment/enjoyment of an educational activity and subsequent student engagement with the content. This finding indicates that students are at least engaged with the content, which can be an important prerequisite for its effectiveness.

A linear regression model regressing self-reported benefit on the control variables presented in Table 6 showed that self-reported benefit was larger among females ($b = 0.12$, SE = 0.04, $p = 0.007$), attendees from the provinces ($b = 0.23$, SE = 0.04, $p = 0.000$), and those who had previously been exposed to extremism ($b = 0.33$, SE = 0.08, $p = 0.000$). Attendees' age did not affect self-reported benefit.

**Table 6.** Covariates of self-reported benefit from show.

|  | Model I |
|---|---|
| Gender (reference = male) | 0.12 (0.04) ** |
| Age | 0.02 (0.01) |
| Location (reference = city) | 0.23 (0.04) *** |
| Exposure to extremism | 0.33 (0.08) *** |
| Constant | 3.36 (0.08) |
| Adj $R^2$ | 0.030 |

Note: $N$ = 1680. Self-reported benefit from show is scaled 0-1. Exposure to extremism is scaled 0-1. Standard errors in parentheses. $^\dagger p < 0.10$, $^*p < 0.05$, $^{**}p < 0.01$, $^{***}p < 0.001$.

The finding regarding previous exposure to extremism is particularly interesting. A total of 39% of participants had experience with somebody they know "liking" extremist content online, 26% had experienced extremist propaganda, either at school or in their neighborhood, and almost a third (31%) had experienced having concerns about extremism in relation to someone they know personally. Exposure to extremism was significantly higher amongst participants in cities than amongst participants from the provinces ($b = -0.07$, SE = 0.01, $p = 0.000$), while there was no difference in exposure across gender and age. The findings thus suggest that authorities seem to be targeting the "right" people, and that this group is also receptive to the message.

## 6. Discussion

Beyond adding empirical depth through the largest analysis to date of the effects of using former extremists to build resilience to radicalization, the findings make four major contributions to the literature. Firstly, they enhance the limited body of research exploring the use of former extremists in P/CVE. This literature, which is largely built on theoretical assumptions, suggests that the use of formers can have positive effects [21,27]. Analysis of the Danish data presented here identified the effects of the project on increasing attendees' ability to recognize extremist ideas and recruitment methods, significantly reduced the proportion of students who perceived political violence as acceptable, and increased attendees' confidence in knowing what to do if exposed to extremism and how to access support. These empirical data support the assumptions behind existing practices that utilizing former extremists can be a powerful tool in efforts to prevent and counter radicalization [78]. That attendees themselves rated the project so highly also indicates that this is a format that students find engaging and that the formers have credibility with attendees. Whilst further research is required, the findings strengthen the existing view that former extremists can play an important and unique role in P/CVE.

Secondly, there were also negative consequences of exposing attendees to the experiences of former extremists—primarily a reduction in political tolerance. Whilst this provides some evidence in support of the wide range of scholars who have focused on the unintended or counterproductive consequences of P/CVE [79,80], it aligns more closely with research that has identified mixed effects of P/CVE projects. As noted, Feddes, Mann, and Doosje's evaluation of a P/CVE project working with Muslim adolescents to increase resilience to radicalization found that the training was largely successful in achieving its aims, but had the unintended additional effect of marginally raising narcissism [31]. The findings of this paper, building on the results of other studies that identified mixed effects, indicate the need for debates about, and design of, P/CVE to evolve beyond the dichotomous arguments between those who present P/CVE work as essential and those who criticize it for having negative effects. Instead, a cost–benefit analysis approach to the design and evaluation of P/CVE is necessary, especially where projects seek to influence attitudes. Those delivering P/CVE need to consider and mitigate potential negative effects in addition to the traditional priority of maximizing benefits, whilst academics—especially those who are critical of such programs—need to consider a broader impact of P/CVE. Such an approach will add maturity to project design and academic discourse.

Thirdly, findings build on studies from neighboring policy areas about the effects of using formers for a range of prevention initiatives. Whilst the findings presented in this paper identify some positive effects, the negative effects reinforce a strand in the formers literature that identifies the risks. As noted, using criminals to prevent youth delinquency has, in some cases, had harmful effects [48]. The drop in political tolerance identified in this paper highlights these risks across policy areas and the importance of practitioners balancing the potential positives and negatives of using formers more generally to address social problems.

Finally, in terms of P/CVE practice, the findings suggest that state-funded P/CVE projects can have significant positive effects in mitigating some of the key drivers of violent extremism. This is especially significant in a youth sample, considering the specific vulnerabilities for this cohort. Additionally, whilst this paper assessed only one project in one country, the analysis nevertheless indicates that the data used by authorities to identify vulnerable cohorts to take part in P/CVE projects appear effective. Comparison of baseline attitudes of project attendees (control group) with the supplementary survey conducted by the authors with a sample of 658 Danish high school students (representative on region and gender) found that, on average, participants attending the formers project had lower political tolerance (M = 0.68 versus M = 0.77, scale 0–1), lower confidence in their ability to recognize extremist ideas (M = 3.69 versus M = 3.87, scale 1–5), higher acceptance of political violence against public authorities or politicians (33.5% versus 5.0%), and higher previous exposure to extremism (M = 0.22 versus 0.14, scale 0–1). These findings thus suggest that authorities seem to be targeting the "right" people and highlight that prioritization of students in larger cities will be especially important in the future.

*Limitations and Future Research*

This study faced several practical limitations, and additional research is required to understand the effects of using former extremists in P/CVE projects better. Limitations stem in large part from the scale of the study and the limited number of survey questions. Whilst this is the largest study to date of the effects of using formers with young people as a P/CVE intervention, it provides only a snapshot of immediate effects. It can say little about the longevity of the intervention's effects, which is essential to long-term strategies for addressing violent extremism. Similarly, perceived legitimacy of political violence findings relate to one single item (violence against public authorities or politicians). There were three methodological and practical reasons for using this item, Firstly, the use of an indirect list experiment to access attitudes allows for only one critical item. Secondly, this was a scenario of particular interest to the project funders, who are concerned about direct threats to democracy. Finally, the item had been used and validated in the baseline study of 658 Danish high school students, and thus reusing the item allowed for direct comparison between samples. Nevertheless, it provides only a limited assessment of the effects of the project on perceived legitimacy of political violence, and further research is required to assess other scenarios to strengthen confidence in the effect.

A further limitation is that whilst statistically significant effects were identified, effect sizes were relatively small for most outcomes. Small effect sizes are, however, to be expected from a one-off intervention of this kind lasting an hour and a half. Finally, the onerous data collection process and the limited time the project organizers were able to dedicate to evaluation during the project meant that only a relatively small set of survey questions could be included. Previous research has highlighted the potential negative effects of P/CVE interventions, and the researchers would have liked to have tested for further potential negative effects.

Three strands of future research would be particularly fruitful. The first is to investigate any differences in project effects between attendees identified as particularly vulnerable compared to other attendees. This requires more detailed data than was possible to collect in this study. Nevertheless, several indicators in the current dataset suggest that this will be an important avenue of inquiry. Using a variable where the funders identified a subset of performances at which the attendees were considered to be particularly vulnerable to radicalization (the number of students in these performances was too small to produce results that we can be confident are reliable and the variable was therefore excluded in the analysis above), we could see a positive and significant interaction between being identified as especially vulnerable and previous exposure to extremism in predicting benefit from the performance. In other words, vulnerable individuals who have experienced exposure to extremism appear to benefit more from the project, and there seems to be a large potential gain from targeting vulnerable students.

A second important research challenge, noted above, will be to assess the lasting effects of the project. A follow-up analysis 6 or 12 months later would allow researchers to investigate the degree to which effects last over time, and whether this varies by sub-group (e.g., younger attendees, city residents, gender). This is especially important in light of analyses of other counter-terrorism campaigns that found that effects, whilst remaining higher than the original baseline, had reduced significantly within 12 months [81]. Analysis of this question can inform practical policy delivery in terms of project renewal and scale. Whether the type of former makes a difference (e.g., age, gender, ideology) and how that can be optimized to different audiences is another important question, as is whether the drop-off in political tolerance is limited to violent groups (i.e., the stories told on stage) or extends to non-violent extremist groups. Finally, qualitative research exploring issues raised by the quantitative analysis would nuance our understanding of the themes that emerged. In particular, it would be valuable to explore with attendees the mechanisms through which the show transforms perceptions and shapes resilience and to explore the experiences of exposure to extremism (e.g., where, by who, through what mode). This is particularly so for experiences of friends "liking" extremist content online, considering the increased prevalence of impactful and sophisticated online propaganda by extremist groups [82].

## 7. Conclusions

This article has presented findings from the largest analysis to date of the effects of using former extremists to build public resilience against violent extremism. Based on a randomized controlled effect evaluation to assess the Danish P/CVE initiative against the aims identified by those funding and delivering the project, findings show mixed effects. The project proved effective in increasing the ability of attendees to recognize extremist ideas and recruitment methods and in increasing confidence in knowing what to do if exposed to extremism and how to access help. The project also appears effective in reducing the perceived legitimacy of political violence against public authorities or politicians, although further research assessing further aspects of political violence is necessary. However, the authors found no effects in relation to political efficacy, whilst political tolerance was slightly lowered by exposure to the project.

The evaluation thus provides support to the principally theoretical claims in the literature that former extremists can be credible and effective partners for P/CVE delivery. However, the mixed effects are an important reminder of the risks of unintended consequences from P/CVE initiatives, especially those seeking to influence attitudes. A cost–benefit analysis framework for designing and evaluating P/CVE projects is necessary for both practitioners and academics to move forward P/CVE debates and evaluation practices. The results also align with literature from neighboring policy areas, such as crime prevention (e.g., Scared Straight), that there are potential risks associated with using formers and these need to be weighed against the potential benefits.

**Author Contributions:** Conceptualization, D.P. and L.L.; Data curation, D.P.; Formal analysis, L.L.; Funding acquisition, D.P.; Investigation, D.P.; Methodology, L.L.; Project administration, D.P.; Resources, L.L.; Supervision, L.L.; Writing—original draft, D.P.; Writing—review and editing, L.L. All authors have read and agreed the published version of the manuscript.

**Funding:** This research was funded by H2020 Marie Skłodowska-Curie Actions, grant number 750348. This research was funded by Aarhus University Research Foundation, grant number AUFF-E-2015-FLS-8-3.

**Conflicts of Interest:** The authors declare no conflict of interest.

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
