# Peer review of "Preventing Extremism with Extremists: A Double-Edged Sword? An Analysis of the Impact of Using Former Extremists in Danish Schools"

_education, doi:10.3390/educsci10040111_

Round 1

Reviewer 1 Report

This is a very high quality paper that will make an important contribution to the field. It is very clearly written and structured. I recommend that it be published, and offer only a few minor suggestions:

* In the Abstract, the authors note that the lack of evaluation to date is surprising given the recognised risks and use formers in other areas. I think this the lack of evaluation is not that surprising, given the relative lack of empirical research in terrorism studies as a whole (which they discuss) and the frequent difficulties in gaining access to data for national security reasons etc as the literature often notes. In this case, involving former extremists in school settings would be more risky than former petty criminals, so this is understandable. Not hugely important to change but could be reworded.
* A little more on the use of former extremists could be included in the introduction - e.g. the context from the para starting on line 103 could be moved up into the introduction.
* What is the definition of a former extremist? There is mention of different ideologies, but given that definitions of extremism differ quite widely (violent vs non-violent, ‘contrary to fundamental values’ or something more, etc), is this category necessarily so assumed or homogeneous? Also as the levels of previous involvement in a group can differ significantly - e.g. those on the fringes vs those directing activities. Could some brief critical discussion of the idea of this category be offered?
* Line 65 ’The process of radicalisation has been widely explored’ - could a brief explanation of the field be added here rather than just the references? I know there would be a lot to cover, but at least a brief definition of what radicalisation means and some of the factors that contribute to it?
* A few words could be added to the explanation of the Danish cartoon crisis (line 150) for an international audience - ‘cartoon crisis, in which images depicting the Prophet Muhammed…etc’
* The effect sizes are small so further studies would help here, but the authors acknowledge this and don’t overstate their conclusions - overall a highly accomplished piece.

Reviewer 2 Report

This paper addresses an important issue, namely the lack of evaluations related to CVE and similar projects. In general, I find the paper interesting, well written and contributing to the field. However, I have some minor remarks, considering smaller re-writing/re-structuring and to include some research concerning radicalization processes which will improve the quality of the paper.

Firstly, the last paragraph in the introduction needs to be more precise and clearer to guide the reader. Now it is formulated as an abstract (addressing the gap, presenting result, contribution and impact), but I think that the paper will be stronger if the author(s) first presented the research gap, then presented the aim (testing the goals). Thereafter, the disposition or structure of the article can be presented. As it is now, it is (at least for me) a bit confusing to read, and also if the major results are presented early – why should the reader read the full paper at all?

Section 2 (on evaluating) needs a slight re-structuring to be clearer, mainly the two first paragraphs. I suggest that the author(s) try to sort out what to say and in what order, f.e. identifying the gap and recent advances needs to be more integrated, and there is also a need to discuss if it is a lack of (quantitative) evaluations or if it is poor data quality that is the main problem. Also, some of the results from previous research needs to be addressed, f.e. the marginal increase in participants narcissism.

In section 4 (Methodology) I miss some words about the time-period, since the pre- and post-evaluation is conducted in relation to the intervention. Even if the author(s) address this in the limitations, it will be good to mention it also in the methodology section, to make the study design more transparent.

Secondly, I miss a discussion on (self-)radicalization and especially which kind of mechanisms the intervention is targeting. Thus, if the major cause to radicalization is considered to be on an individual level, this kind of interventions might have some effect, but if it is process oriented the intervention needs to target exposure or interaction to be effective (see Crettiez 2016; Smith 2018; Spaaij 2012). I think it is crucial to address the link between the theory or mechanism that cause radicalization and explain how the intervention will target that mechanism. I suggest this can be done in section 3 (to contextualize the theory) or in section 1 (to address it in general).
